# Navigating Intercultural Medical Encounters: An Examination of Patient-Centered Communication Practices with Italian and Foreign Cancer Patients Living in Italy

**DOI:** 10.3390/cancers15113008

**Published:** 2023-05-31

**Authors:** Filomena Marino, Francesca Alby, Cristina Zucchermaglio, Teresa Gloria Scalisi, Marco Lauriola

**Affiliations:** Department of Social and Developmental Psychology, Sapienza Università di Roma, 00185 Roma, Italy; filomena.marino@uniroma1.it (F.M.); francesca.alby@uniroma1.it (F.A.); cristina.zucchermaglio@uniroma1.it (C.Z.); gloria.scalisi@uniroma1.it (T.G.S.)

**Keywords:** patient-centered communication, oncology encounters, intercultural communication, ONCode coding system, video-recorded visits, communication barriers, cancer care quality

## Abstract

**Simple Summary:**

Good communication is key in cancer care, especially when doctors and patients come from different cultures or speak different languages. We studied 42 videos of doctors talking to Italian and foreign cancer patients during their visits. We looked at how they talked to each other, whether they misunderstood anything, whether there were interruptions, and how much trust and emotion were shown. The type of appointment and the doctor’s personal style mattered more than whether the patient was Italian or foreign. This tells us that even when foreign patients can speak the language well, doctors cannot only rely on this to communicate effectively. Doctors should pay attention to interruptions and focus on taking care of the patient as a whole person. The methods we used in this study could help doctors improve their communication skills, which will lead to better care for all patients.

**Abstract:**

Effective communication is crucial in cancer care due to the sensitive nature of the information and the psychosocial impact on patients and their families. Patient-centered communication (PCC) is the gold standard for providing quality cancer care, as it improves patient satisfaction, treatment adherence, clinical outcomes, and overall quality of life. However, doctor–patient communication can be complicated by ethnic, linguistic, and cultural differences. This study employed the ONCode coding system to investigate PCC practices in oncological visits (doctor’s communicative behavior, patient’s initiatives, misalignments, interruptions, accountability, and expressions of trust in participants’ talk, Markers of uncertainty in doctor’s talk, markers of emotions in doctor’s talk). Forty-two video-recorded patient–oncologist encounters (with 22 Italian and 20 foreign patients), including both first and follow-up visits, were analyzed. Three discriminant analyses were conducted to assess differences in PCC between patient groups (Italian or foreign patients) according to the type of encounter (first visit or follow-up) and the presence or absence of companions during the encounters. Multiple regression analyses were performed to evaluate the PCC differences by oncologist age, patient age, and patient sex, controlling for the type of encounter, the presence of a companion during the visit, and patient group on ONCode dimensions. No differences were found in PCC by patient group in discriminant analyses and regressions. Doctor communication behavior, interruptions, accountability, and expressions of trust were higher in first visits than in follow-ups. The disparities in PCC were primarily linked to the type of visit and the age of the oncologist. However, a qualitative analysis showed notable differences in the types of interruptions during visits with foreign patients compared to Italian patients. It is essential to minimize interruptions during intercultural encounters to foster a more respectful and conducive environment for patients. Furthermore, even when foreign patients demonstrate sufficient linguistic competence, healthcare providers should not solely rely on this factor to ensure effective communication and quality care.

## 1. Introduction

Oncology visits offer a unique ecology for studying communication and its effects. The unique characteristics of cancer care influence the communication process between doctors and patients. Neoplasms are serious and potentially life-threatening diseases, but they also hold the possibility of cure. Therapies involve numerous treatment modalities and different professionals (oncologists, surgeons, radiotherapists, nurses, etc.). Furthermore, during these encounters, high stress levels, uncertainty, complex information, and life-changing medical decisions are involved. During oncology visits, physicians and patients engage in an intense communication activity that involves providing, understanding, and remembering information about the disease and treatments. Effective communication can alleviate suffering by improving the emotional well-being of patients. It can also indirectly improve treatment adherence [1].

Recognizing the importance of effective communication in oncology, ASCO [2] published guidelines for physicians on Patient-Centered Communication practices (PCC). PCC is a multidimensional construct that involves the co-construction and integration of instrumental, affective, and participation behaviors [1,3,4,5]. PCC aims to improve the effective exchange of information between participants, respond to the emotions expressed by the patient, and manage the uncertainty of the encounter. Moreover, PCC contributes to building a lasting relationship between the parties and involves the patient in the care process and therapeutic decisions through discussion and shared understanding of the disease and available treatments [1,6,7,8,9,10].

The importance of PCC practices is reflected in studies that demonstrate its positive impact on patient well-being. It is well established that PCC can have a positive influence on post-visit health outcomes, including satisfaction, reduced emotional distress, anxiety, and improved physical and psychological quality of life [6,11,12,13]. In addition, patients who are more involved in the decision-making process tend to be more satisfied with the encounter. They are more knowledgeable about their disease, exhibit higher treatment compliance, are better able to control their condition, and experience an improved quality of life after diagnosis and treatment [14,15]. Finally, empathic listening, addressing patients’ concerns and fears, and providing reassurance all contribute to the establishment of a trusting relationship. This, in turn, leads to greater compliance with medical treatment, better psychological health, reduced emotional distress, and decreased anxiety [16,17,18,19].

Providing a diagnosis and clear, written information about treatment, giving space for the patient to ask questions, seek clarification and express doubts, as well as accepting the patient’s concerns and emotional states are communicative activities that, along with other actions such as: explaining and discussing the severity of the disease, providing comprehensive information about the available options and the treatment pathway, would define lower levels of cancer-related anxiety and depression, higher levels of satisfaction following the encounter, and greater competence, compliance, adherence, and self-management of the proposed therapy [4,20,21]. In turn, adherence to treatment increases the chances of survival, enables better management and quality of therapeutic outcomes and, from a broader perspective, leads to a reduction in healthcare costs and unnecessary treatments [1,7,19,22,23,24,25,26,27,28,29,30,31].

In this sense, there is a “virtuous circle of PCC” [32] in which the more the oncologist integrates and co-constructs a PCC with patients and caregivers during visits characterized by affective, instrumental, and participative behaviors, the greater the chances of establishing a solid relationship of trust between the parties; that the patient will have a better understanding of the illness and the therapies that the patient will comply with the proposed medical treatments, and that there will be positive results on his or her health and psychological well-being over time.

The virtuous circle of PCC tends to be challenged when communication takes place between doctors and foreign patients. Studies conducted in an intercultural context illustrated the complexity of doctor–patient communication when the latter is a foreigner and there are linguistic, communicative, interpretative, and relational barriers [33]. Studies of migrant patients have shown that they face multiple sources of stress when engaging with healthcare, including difficulties in understanding or navigating the local healthcare system, communication challenges, and language barriers. Other factors that contributed to inequalities in health outcomes among foreign patients included an inaccurate understanding of diagnosis and treatment and an increased incidence of communicative misunderstandings regarding the causes of cancer. These barriers, in turn, affect the comprehension of diagnosis and treatment, resulting in poor long-term health outcomes [34,35,36,37,38,39,40]. Such barriers are also associated with more significant psychological distress and lower quality of life among foreign cancer patients [41,42,43]. Furthermore, the different linguistic, ethnic, and cultural identities of patients and physicians influenced the expectation of mutual understanding [39,44,45]. On the one hand, foreign patients exhibit less assertiveness and verbal expressiveness and experience difficulties in using effective conversation strategies. On the other hand, physicians showed greater emotional detachment, adopted more directive behavior, and spent less time explaining the disease and treatment to foreign patients.

The few studies investigating PCC in the intercultural context underline physicians’ difficulties in building patient-centered interactions and the communication needs of foreign patients during medical visits. Research with migrant women highlighted barriers to the implementation of PCC due to language, the poor training of doctors, and organizational difficulties within hospitals and the health care system. Research also highlighted the need for doctors to have more knowledge about how to establish a PCC with migrant patients and the lack of attention given by health professionals to their concerns and questions. These patients also emphasized the need for more information and time to discuss their condition with professionals [46]. Studies conducted in the US on PPC in ethnic minorities showed that compared to white men, black and Hispanic/Latino patients reported fewer experiences of patient-centered medical visits, greater unmet needs, more significant experiences of discrimination, and dissatisfaction with treatment discussions [47]. Similarly, studies conducted in the US and Canada with Asian-American and Asian-Canadian cancer patients (male and female) reported that they were less likely to be involved in treatment decisions compared to white patients, and language was the main factor contributing to perceived disparities [47,48,49,50,51]. On the other hand, studies investigating the positive effects of PCC on the experiences of migrant patients showed that satisfaction with the care received increased, regardless of the racial concordance between the doctor and the patient. This increase was observed when the specialist spent time with the patient, informed and explained the patient’s health condition, showed support, and involved the patient in treatment decisions [52,53].

Drawing on a corpus of video recording of oncology visits, the present study aims to contribute to the literature by exploring PCC practices in the Italian medical context. Italian medical context is understudied in relation to PCC practices with ethnically diverse groups. Our first objective is to explore potential differences in PCC practices in two kinds of medical encounters: oncology visits between Italian doctors and Italian patients and oncology visits between Italian doctors and foreign patients. Our second objective is to verify if the differences in PCC depend on the type of visit and the presence/absence of companions. Finally, we analyzed the effects on the PCC of six non-interacting variables, including the type of encounter, age of the oncologist, age of the patient, patient’s gender, the presence of a companion, and the nationality of the patient.

## 2. Materials and Methods

### 2.1. Participants

The video-recorded medical visits analyzed in this study were collected in the oncology unit of a medium-sized Italian hospital operated by a Religious Order during ethnographic research in 2019. The research received approval from the Ethics Committee of the hospital (Prot. No. 1886/CE Lazio) and the Ethics Committee of the Department of Social and Developmental Psychology, Sapienza University of Rome (Prot. No. 0000944). These videos included interactions between eight different oncologists (equally distributed by gender) and 42 patients, of which 22 were Italian (17 women, age M = 57 years; 5 men, M age = 71) and 20 foreigners (15 women, M age = 52 years; 5 men, M age = 61). The foreign patients’ countries of birth were Romania, Peru, Albania, USA, Moldova, Cameroon, Bulgaria, Ecuador, Kosovo, Egypt, Poland, the Philippines, Ukraine, Mexico, and Paraguay. The heterogeneity of the ethnic background of these patients reflects the inhomogeneity of the Italian context. According to census data [54], foreigners residing in Italy as of 1 January 2022, number 5,030,716 and account for 8.5 percent of the total population. Except for Romanians, who account for about 20% of all foreign residents, there are four groups each representing 5–10%, and 12 groups each representing between 1 and 5%. Friends or relatives accompanied eleven Italian and nine foreign patients in the examination room. The most common neoplasms among the participants were breast cancer (42.9%), followed by gastrointestinal stromal tumors (21.4%), gynecological cancers (11.9%), lung cancer (11.9%) and head and neck cancer, urological cancers, neuroendocrine tumors, liver cancer and lymphoma tumors (2.4% each). Italian language proficiency was assessed retrospectively to avoid selection bias. Specifically, video-recorded visits were coded using the indicators included in the Common European Reference Framework for Languages [55], and a language proficiency score (encompassing both comprehension and production) was assigned to each patient. Ninety percent of the patients demonstrated at least a sufficient knowledge of the Italian language, and 65% of them were highly proficient. Generally, patients with lower competence in the Italian language were those who were accompanied most frequently.

### 2.2. Materials and Procedure

This corpus consisted of 42 video recordings, selected to represent a diverse range of encounters, such as follows: (a) first-time, post-surgical encounters; (b) follow-up oncological visits; (c) intercultural oncological encounters with foreign patients; (d) oncological visits with Italian patients. By including diverse encounters, our goal was to provide comprehensive coverage of communication dynamics in different contexts within the oncology setting. We collected data during the first time post-surgical visits (20 in total, equally distributed by foreign and Italian patients) and follow-up visits (22 in total, ten with foreign patients and twelve with Italian patients). Oncologists, patients, and companions were invited to participate in the study while waiting for their appointment. Those who agreed to participate were asked to sign a written informed consent form to take part in the study and allow video recording of the visit. A video camera was placed in the examination room, focusing on the patients, companions, and physicians interacting.

We used the ONCode tool to assess PCC during over 13,000 interaction turns, amounting to a total of 18 h of oncological visits. ONCode is a new coding system specifically developed to capture doctor–patient communication practices in post-surgical oncological encounters [56]. It was developed based on video-recorded consultations in Italian hospitals from 2012 to 2020. ONCode is grounded in emic notions of PCC, together with evidence and definitions provided by the literature [57,58,59]. Moreover, previous studies provided the ground for a situated definition of the PCC dimensions used in ONCode. Prior work provided an empirical ground for understanding how patient-centered communication unfolds within the cultural, organizational, and medical constraints of oncological consultations in Italy [60,61,62,63,64] while also documenting the overall structural organization of these oncology consultations, which routinely include a sequence of stages and activities [65,66,67,68]. By examining the extent to which and how each participant (i.e., doctor, patient, and companion, if present) participated in various communicative actions, ONCode provides the following interactive dimension scores: physician’s communicative behavior (DCB), patient’s initiatives (PI), misalignments between doctor and patient (MIS), interruptions in visits (INT), accountability and expressions of trust in the talk (ACC), markers of uncertainty in doctor’s talk (MOU), and markers of emotions in doctor’s talk (MOE). In a previous study [56], ONCode scores displayed good reliability even when employed by a single observer and had incremental validity above other existing coding systems for analyzing PCC. Table 1 reports a thorough description of each score, including operative definitions and examples. In the present study, we also considered the following non-interactional variables as potentially affecting PCC: the age of the oncologist, patient’s age, patient’s sex, type of encounter, presence of a companion during the visit, and patient’s nationality.

### 2.3. Statistical Analysis

The skewness and kurtosis statistics of the seven ONCode scores ranged between −2 and +2, suggesting that the assumption of normality has not been violated, and parametric statistics can be applied [69]. Furthermore, the presence of heteroscedasticity was checked, and it was found that all the residuals were homoscedastic. To determine whether there are differences in PCC between visits with Italian and foreign patients, a discriminant analysis was carried out on the seven ONCode scores, considered simultaneously. Similarly, to examine whether there were differences in the PCC within the corpus based on the type of visit and the presence of the companion, two additional discriminant analyses were carried out on the seven ONCode interactive dimensions, considering them simultaneously. Finally, to analyze the effect of the noninteractive variables (i.e., age of the oncologist, patient’s age, patient’s sex, type of encounter, presence of companion during the visit, and patient’s nationality) on the ONCode interactive dimensions, seven separate multiple standard regressions were performed for each interactive dimension of the system, while ruling out multicollinearity issues. All of the analyses were performed using STATISTICA 13.

## 3. Results

### 3.1. Discriminant Function Analyses

In the first discriminant analysis, the investigation focused on examining the presence of differences in PCC between the Italian patient group and the foreign patient group using the patient’s nationality as the group factor. However, Fisher’s F value was found to be nonsignificant, F (7, 34) = 0.446, *p* = 0.86. The seven ONCode scores did not discriminate between the groups, nor did they make a unique significant contribution to the discrimination between the groups. Thus, no differences in PCC were found during visits depending on whether the patients belonged to the Italian or foreign group of patients.

In the second discriminant analysis, the presence or absence of the accompanying person considered during the visits was considered the grouping factor. The main results are reported in Table 2. The overall model was not statistically significant in discriminating between the groups based on the presence of a companion, F (7, 34) = 1.62, *p* < 0.16. A closer examination of the individual variables showed that none of them individually contributed significantly to the discrimination between the groups, except for interruptions (INT) with partial Lambda of 0.84, *p* < 0.01. The frequency of interruptions played a role in differentiating between the groups, and this variable’s contribution to distinguishing unaccompanied patients from accompanied ones was independent of any contributions made by other variables.

As shown in Table 3, the number of interruptions during visits increased when the patient was accompanied.

The last discriminant analysis considered the type of encounter, comparing the group of first visits with the group of follow-up visits (Table 4). The significant omnibus test, F (7, 34) = 3.17, *p* = 0.01, indicated that the set of variables considered discriminated between the types of encounters. The analysis also revealed that interruptions (INT) had the highest F-remove value (3.18) and the lowest *p*-level (0.08), suggesting that they may be the most important ONCode variable to distinguish between first visits and follow-ups. Other variables, such as doctor’s communicative behavior (DCB), accountability and expressions of trust in participants’ talk (ACC), and markers of uncertainty in doctor’s talk (MOU), also had relatively high F-remove values and low *p*-levels, indicating that they may contribute to the differences between the two groups as well.

Although no individual variable (partial Lambda) reached the conventional levels of statistical significance, the overall analysis of discriminant functions was significant, indicating a significant difference between the two groups when considering all variables collectively. This finding suggests that a multivariate effect may have obscured the univariate effects of the variables, making it challenging to determine the individual contributions of each variable to the differences between the two groups.

Examining the standardized coefficients of the discriminant function reported in Table 5 can be an appropriate analytic approach to identify the variables that contributed the most to discrimination between first visits and follow-ups within the context of a significant multivariate effect. This approach allowed us to better understand the complexity of the multivariate effect and gain insights into the relative importance of each variable in differentiating PCC by type of encounter.

Using a cut-off point of 0.50 for interpretation, Table 5 revealed that doctor communicative behavior (DCB), interruptions (INT), and accountability and expressions of trust in participants’ talk (ACC) had the strongest positive coefficients, indicating that these variables were more prominent in distinguishing the two groups. The markers of uncertainty in doctor’s talk (MOU) had a moderate positive coefficient, indicating a moderate contribution to the differences between the groups. On the other hand, patient’s initiatives (and companion) (PI), misalignments (MIS), and markers of emotions in doctor’s talk (MOE) had negative coefficients, suggesting that these variables were less influential in differentiating between encounters.

As shown in Table 6, the ONCode dimensions that had the greatest influence on the discriminant function (i.e., doctor communication behavior, interruptions, accountability, and expressions of trust) were higher in first-visit encounters. Similarly, markers of uncertainty were higher during first-visit encounters, but this ONCode dimension had a lower weight on the discriminant function.

Anticipating the discussion, more interruptions occurred during initial oncology visits than during follow-up visits, possibly due to the longer duration of first-time encounters or the increased participation of companions. Furthermore, oncologists used more PCC strategies during initial consultations than during follow-up visits. This was also related to the presence of trust and responsibility expressions in the conversations. In fact, during the initial appointments, doctors provided diagnostic explanations that supported the choice of medical treatments that patients would need to follow over time.

The ONCode scores were able to correctly classify 82% of the follow-up visits, while first visits were correctly classified in 75% of the cases. This result underscores the strong discriminatory ability of all dimensions of the ONCode in distinguishing between types of visits. Additionally, the square of the canonical correlation coefficient (Canonical R^2^ = 0.39; Chi^2^ = 18.32, df = 7, *p* < 0.05) indicated that the set of variables explained approximately 40% of the variance in the PCC due to differences in the type of encounters.

### 3.2. Regression Analyses

We conducted seven multiple standard regression analyses, one for each dimension of the ONCode, using the non-interactive variables collected in the present study as predictors: oncologist age, patient age, patient sex, type of interaction, presence of a companion during the visit, and patient nationality. The only significant analysis was for interruptions (INT) as the dependent variable. As shown in Table 7, there was a significant relationship between interruptions and several variables, namely the age of the oncologist, the nationality of the participant, and the presence of the companion. These variables significantly contributed to the prediction of interruptions, R^2^ = 0.37, *p* < 0.01, and F (6, 35) = 3.47. The beta coefficient for each variable was positive, indicating that higher values on these variables were associated with a higher frequency of interruptions. Specifically, older oncologists (β = 0.44), patients from foreign countries (β = 0.31), and the presence of a companion during the visit (β = 0.49) were associated with more interruptions during the medical encounters. The other variables, including the type of encounter, patient’s age, and patient’s sex, did not significantly contribute to the prediction of interruptions.

All independent variables had relatively low levels of redundancy, as shown in Table 8, and each made a unique contribution to the prediction of interruptions.

Since the beta values of the three variables were significant, we can compare their unique contributions by squaring the semipartial correlations shown in Table 9. The results indicated that the presence of a companion had the highest semi-partial correlation (0.44), followed by oncologist age (0.39) and patient nationality (0.29). These variables were the most important predictors of oncological visits, and there were no collinearity issues in the analysis.

It should be noted that in the discriminant analyses, the ONCode variables (including interruptions) did not differentiate patients by nationality. However, in the regression, belonging to the Italian or foreign patient group appears to be a significant predictor of the “interruptions” variable. This discrepancy is likely due to the contribution of the other variables included in the regression, such as the patient’s age and gender, which may have acted as “suppressors” [70] of the variance related to nationality that was not directly associated with the “interruptions” variable. As a result, there was an increase in the component of “unique” variance shared between nationality and interruptions.

### 3.3. Qualitative Analyses

Prompted by regression analysis, we conducted a qualitative analysis of the interruption occurrences to gain a deeper understanding of the interruptions that occurred throughout the encounters. Our analysis revealed that oncologists aged over 40 experienced interruptions more frequently than their younger counterparts. The interruptions experienced by older oncologists included requests for advice, comments, and comparisons from peers in the same specialty about other patients, consultations from other specialists (such as radiotherapists) regarding ongoing treatments for shared patients, the secretary entering the room to ask for information and/or resolve issues with patients and/or appointments, the secretary entering the room to collect or return medical records, a patient leaving the room. The head of the department, being one of the senior oncologists, experienced the most interruptions during visits. These interruptions were not only for the reasons mentioned above but also due to the need to address issues related to the management and organization of the oncology department. The “nature” of the interruptions, especially those involving requests from other coworkers (such as seeking advice on another patient’s ongoing treatment or seeking information on a specific case) and/or consultations from other specialists (such as discussing the possibility of performing radiotherapy on a patient), was a practice that, in this corpus, was observed only in the visits of experienced oncologists with several years of clinical practice and who had been collaborating with other oncologists for an extended period. The consultations with younger oncologists (under 40 years old) were occasionally interrupted and typically for non-organizational reasons, such as the secretary entering the room or the doctor’s mobile phone ringing.

To gain a better understanding of the reasons for interruptions during visits with foreign patients, we focused on distinguishing between different sources of interruptions. Specifically, we categorized interruptions into two groups: those originating from outside the examination room (e.g., the secretary knocking on the door or phone ringing) and those originating from inside the room (e.g., the doctor leaving the room or making unrelated phone calls during the consultation). Our findings indicated that interruptions during visits with Italian patients were primarily attributed to external events that involved only the oncologist or hospital staff. However, in four instances during intercultural visits, the interruption of the visit could be attributed to phone calls made to the private mobile phones of patients and caregivers. Additionally, visits with foreign patients experienced interruptions due to internal events that originated from the oncologists themselves. For instance, on six occasions, the oncologist made phone calls for professional or organizational activities that were unrelated to the ongoing visit. Thus, in contrast to visits with Italian patients, visits with foreign patients were more susceptible to interruptions caused by both external events affecting patients and their companions as well as internal events initiated by the oncologist.

## 4. Discussion

Using a corpus of video recordings of oncology visits and a recently developed observational coding system (ONCode), the present study aimed to explore PCC practices with cancer patients in an Italian religious-order-operated hospital. Our first objective was to investigate differences in PCC practices in medical encounters between Italian doctors and Italian patients and those involving Italian doctors and foreign patients. Previous research addressing this issue has reported differences in communication practices during intercultural encounters with foreign patients, highlighting the role of linguistic, cultural, and ethnic backgrounds [33,34,35,36,37,38,39,40]. Language has been identified as a primary contributing factor to difficulties in PCC [48,49,50,51]. However, doctor training and organizational challenges within hospitals and healthcare systems have also been reported as barriers to effective PCC [46,48,49,50,51]. For example, foreign patients have expressed the need for more information and time to discuss their condition with healthcare professionals rather than complaining about linguistic problems [46]. Furthermore, studies conducted internationally have shown that patients from ethnic minorities typically report fewer patient-centered medical visits and more experiences of discrimination compared to white men. For example, qualitative analyses in oncology indicated that black and Hispanic men had more unmet needs during interactions with physicians, received fewer inquiries about their preferences and priorities, and expressed greater dissatisfaction with long-term treatment discussions [47,71,72].

Our research found limited evidence of differences in PCC practices between medical encounters with Italian and foreign patients. In fact, we observed that factors such as nationality and ethnic background had less influence on PCC practices compared to other interactive and non-interactive variables. This contrasts with the findings of most international studies that have emphasized the impact of nationality and ethnic background on PCC practices [33,34,35,36,37,38,39,40,41,42,43]. However, it is important to note that implementing PCC has the potential to bridge the racial and cultural differences between doctors and patients, and it can have a positive impact not only on both native and foreign patients. In this vein, our results align with previous research that has highlighted the positive effects of PCC on patients with a migrant background in the US [46,52,53]. Paraphrasing Chu et al. [53], PCC “is key to reducing disparities and improving immigrant patients’ satisfaction level with medical care”.

In addition to that, it is critical to recognize that our study was conducted within a specific context with its own unique constraints, and further research is needed to gain a comprehensive understanding of PCC in intercultural contacts in Italy. First, the lack of discernible changes between visits by Italian and foreign patients could be attributed to specific features of organizational culture. This religious hospital claims and emphasizes in its mission the centrality of each person in the care process, with particular attention to minority and vulnerable groups, including migrant and foreign patients. This aligns with recent research suggesting that religious hospitals promote a more inclusive and respectful atmosphere, leading to a higher quality of care. This interpretation is consistent with a recent Italian study that examined the differences in the provision of support to cancer patients undergoing chemotherapy between oncologists in a religious hospital and those in a government-operated hospital [73]. Second, it is worth noting that research on PCC with foreign patients in Italy is still in its early stages, and there is a lack of studies specifically examining the impact of language difficulties. While it is possible that linguistic factors may have been less influential in our study, as almost all patients had sufficient knowledge of the Italian language and had good proficiency levels, it is crucial to recognize that patients with good proficiency may still struggle to understand medical terminology in a foreign language. These language barriers can lead to misunderstandings, difficulties in fully comprehending medical information, and a potential lack of trust in healthcare professionals. More research is needed to understand how language difficulties can affect PCC for foreign patients in Italy.

The second objective of the present study was to determine whether PCC was different between first visits and follow-ups or dependent on the presence or absence of an accompanying person. In fact, companions can both facilitate doctor–patient communication and provide support to cancer patients while sometimes being seen as obtrusive, particularly with elderly or vulnerable patients in advanced stages of illness [66,74,75,76]. Therefore, we speculated that the presence of an accompanying person could facilitate intercultural encounters, providing informational and affective support to foreign patients. In our study, however, the presence of an accompanying person and being a foreign patient were found to play a role only in relation to interruptions during the visit. Before discussing this finding, it is worth noting that the oncologist’s age, and thus their role as an organizational senior, was also found to predict PCC categories, and in particular, the interruptions.

In the ONCode framework, interruptions included events such as phone calls, the doctor leaving the room because of matters that did not concern the current consultation and the patient, exchanges with the nurse, or other doctors entering the room. During first-time visits, interruptions occurred more frequently than during follow-up visits. This finding might be attributed to the longer duration of these encounters, which could make them more likely to be interrupted by external events. The fact that the doctor’s age predicted the frequency of interruptions could be attributed to the recognized expertise within the medical team and their greater involvement in organizational matters, a finding that resonates with previous research that considered organizational challenges to be impeding effective PCC [46,48,49,50,51].

To gain further insight into interruptions in terms of PCC practices, we conducted a qualitative analysis to examine the specific nature of interruptions during the visits. Our analysis revealed that encounters with younger oncologists were characterized by sporadic interruptions, such as the secretary entering the room or the doctor’s mobile phone ringing for consultations. In contrast, older oncologists experienced more frequent interruptions due to a variety of reasons, including requests for consultation, expert opinions, and discussions with colleagues in the same specialty, consultations from other specialists, secretaries entering the room for information, problem solving, or medical record collection, and occasionally personal mobile phone ringing. Among senior oncologists, the head of the department experienced the most frequent interruptions, which were often related to additional responsibilities associated with their organizational position and management of the oncology department. These types of interruptions were only observed in oncology visits involving doctors with extensive clinical experience and long-term collaboration with the oncology department.

The finding that older oncologists faced more interruptions could indicate their greater experience and the demand for their advice and consultations. However, it could also imply a lack of adequate organizational support to effectively manage their time and minimize interruptions. Additionally, the observation of interruptions from colleagues and specialists exclusively in visits of oncologists with several years of clinical practice and long-term collaboration with the oncology department raises interesting questions about the influence of professional networks and organizational culture on clinical practice. These insights highlight the need for further research to enhance our understanding of the factors contributing to interruptions in oncology encounters and their implications for patient care.

Understanding why the frequency of interruptions increased in the presence of foreign patients was more challenging. While visits with Italian patients were predominantly interrupted from outside and involved only the oncologist or hospital staff, visits with foreign patients were interrupted from outside but also involved the patient and the companion. In fact, on four occasions, the interruption of the visit can be traced to the telephone calls made to the private mobile phones of patients and companions. Furthermore, on six occasions, it was necessary to interrupt the interaction due to events occurring during the visit; that is, the oncologist interrupted the communicative action with the patient to devote themselves to activities not related to the event in progress. However, on one occasion, the researcher in the room interrupts the visit. Therefore, unlike what happens in visits with Italian patients, it is possible that in the presence of foreign patients, visits are more likely to be interrupted by: (a) external events that also affect patients and their companions, (b) internal events that concern interruptions made by the doctors and sometimes by other participants in the visit. In the first case, contrary to what happened in visits with Italian patients, foreign patients and their companions contributed to interrupting the visit in the same way as the oncologists did: by maintaining contact with the outside world during the interaction and by answering the telephone. In the second case, only in cross-cultural visits did we observe within-site interruptions that originated from the oncologist.

All these findings were unexpected and emerged only after a qualitative analysis. It is possible that in the encounter between an Italian doctor and foreign patients, there are implicit signals on both sides that prime interruptions. Previous research has often interpreted the suspensions of the visit as negative events for PCC [77,78,79]. However, interruptions should not necessarily be seen as negative events. For example, communication interruptions can lead to more patient (or companion) initiatives, such as asking questions or seeking clarification or explanation [80]. The qualitative analysis suggested notable differences in the types of interruptions that occur during visits with foreign patients compared to Italian patients. Interruptions during visits with Italian patients were primarily due to external events caused by hospital staff or the oncologist. On the contrary, visits with foreign patients were disrupted by both external events and internal events initiated by the oncologist. These findings may suggest that foreign patients require additional attention from healthcare professionals to maximize the quality of patient-centered care. If these differences are replicated, an avenue for future research could investigate the antecedents of interruptions in both same the culture and intercultural encounters.

### Limitations

Although this work provided encouraging results on PCC in intercultural medical encounters, it is important to highlight the limitations of our research and the possibilities for future studies to address these problematic knots. The sensitivity of the collected data and the complexity of the context in which the oncological encounters were videotaped determined the first limitation of this study, represented by the limited number of participants. While the limited number of participants and the partial repetition of oncologists’ communication characteristics and behaviors pose research limitations, this data set is valuable, despite the statistical constraint on the number of visits. Future research could leverage larger data sets and ensure enough visits per oncologist. However, it is worth noting that the study’s novelty and importance lie in its analysis of over 13,000 interaction turns from 18 h of oncological visits, providing a rare view of the negotiation and co-construction of PCC in real intercultural interactions. Despite its complexity, such research into PCC as a multidimensional construct is scarce, underlining this study’s value.

Another aspect to be considered is the use of oncologists in the analyses as partially repeated measures. Given that there were eight oncologists, it would be expected that each would show ‘consistent’ behaviors from one visit to the next, thus contributing to the results to some extent. With a larger corpus of available data, each oncologist would have had more visits. It would have been possible to evaluate the effects of the oncologists’ characteristics as clusters, separately from those of the other variables, using multilevel analysis. With these considerations in mind, future research could look at larger datasets with enough visits per oncologist. Finally, the study highlighted the role of interruptions during encounters as opportunities especially exploited by foreign patients. Future qualitative research could explore further which kind of use participants make of such breaks in the ongoing communication during the visits.

## 5. Conclusions

The use of ONCode provided statistically significant results that illustrated how PCC was negotiated and co-constructed between all participants in the visit. In fact, the doctor, the patient, and the companion contributed equally to the development of the oncological encounter. Our findings shed new light on oncology visits and the challenges oncologists must manage daily in clinical practice. These challenges result in concerns about interactional and local features of the unfolding communication event rather than individual patient characteristics (i.e., nationality). Managing the sequential unfolding of the activities of the different types of visits while handling organizational matters and maintaining a patient-centered focus in communication could be quite challenging for physicians.

Based on the findings of the present study, we can suggest some practical implications for healthcare providers to improve PCC in oncology. First, doctors should pay attention to the interruptions during visits, especially with foreign patients, and to the overall management of the unfolding interaction in intercultural encounters. Medical education interventions might target this topic by sustaining doctors’ reflexive awareness of their interactional practices while helping them to create a more conducive and respectful environment for the patient. Second, although linguistic issues have been highlighted as a contributing factor to PCC difficulties, our study discovered that even when foreign patients have enough linguistic competence, healthcare providers should not rely only on this element to assume good communication and quality care. Instead, they should prioritize PCC, especially in international interactions, to maintain patient-centeredness. Third, the finding was that PCC was similar between Italian and foreign patients in the context of a religious-operated hospital. Future studies are needed to verify and further explore this finding that, if confirmed, might contribute to support and promote the humanization of health care.

## Figures and Tables

**Table 1 cancers-15-03008-t001:** ONCode dimensions, operative definitions, and examples.

ONCode Dimension	Operative Definition	Examples
Doctor’s communicative behavior (DCB)	We observed how the doctor accomplished the activities at each stage of the consultation. It concerns affective, instrumental and participation verbal and non-verbal behaviors, which can be present about the activity in progress in each phase.	Some communicative actions included how the doctor recommended a treatment, prescribed the following examinations, or delivered the diagnosis. We relied on linguistic actions, such as questions, meta-pragmatic formulations, explanations, and recommendations. Additionally, we noted whether and how the doctor engaged in small talk, used humor, and allowed the patient to propose questions and initiatives of interest to her or him.
Patient’s initiatives (and companion) (PI)	We observed whether and how the patient and companion co-constructed the encounter.	We examined each patient and companion communication action during each stage of the visit, including whether patients and their companion took initiatives, such as asking questions, expressing concerns, proposing a topic, or simply aligning with what the doctor said or asked.
Misalignments (MIS)	We observed whether a fracture in the co-orientation of participants toward the same activity or in the understanding of the topic had occurred.	What is evaluated is whether, and with what effort, the participants repaired the fracture and reached an agreement or, instead, remained in distant positions.
Interruptions (INT)	We observed interruptions in all the moments in which the consultation was suspended due to matters that did not concern the current consultation and the patient. We checked for interruptions at each stage of the consultation.	Interruptions included phone calls, the doctor leaving the room, exchanges with the nurse, or other doctors entering the room. Interruptions due to systematic organizational routines (e.g., the doctor goes out to photocopy the patient’s exams) were not counted.
Accountability and expressions of trust in participants’ talk (ACC)	We observed how the doctor made her/himself accountable to the patient by providing access to her/his medical knowledge and reasoning. We also assessed if the patient topicalized trust and confidence in the doctor discourse.	The accountability involves explaining the rationale used for recommending treatment, providing alternative options for treatment, and using metapragmatic markers that help the patient orient themselves within the consultation activities. Furthermore, we coded expressions of trust and confidence in patient discourse in the oncology, such as “If the doctor says so, I will do it” or “You are the doctor, and I trust what you say”.
Markers of uncertainty in doctor’s talk (MOU)	We observed mentions of uncertainty in the doctor’s talk, that is, when the doctor showed uncertainty about the treatment outcomes, test results, or treatment possibilities	They refer to all those occasions when the doctor showed uncertainty in their speech by using modalized or evidential moods, reference to probability, emphasizing small benefits, expressing uncertainty of outcomes and test results (e.g., “The recommendation for treatment is not absolute in my opinion, but I tend to prescribe it”).
Markers of emotions in doctor’s talk (MOE)	We observed whether there were occasions during the doctor’s conversation when he/she expressed verbal and non-verbal socio-affective behaviors	We observed whether there were sequences of reassurance (e.g., the doctor highlights positive sides of the situation), jokes or humor, the doctor’s response to emotional concerns expressed by the patient, and whether the doctor gestures of support such as touching hands.

**Table 2 cancers-15-03008-t002:** Discriminant Function Analysis Summary for presence of accompanying person (Unaccompanied Patient vs. accompanied Patient).

N = 42	Discriminant Function Analysis Summary No. of Vars in Model: 7; Grouping: Companion (2 Groups) Wilks’ Lambda: 0.75 Approx. F (7, 34) = 1.62 *p* = 0.164					
	Wilks’ Lambda	Partial Lambda	F-Remove (1, 34)	*p*-Level	Toler.	1-Toler (R-Sqr.)
Doctor’s communicative behavior (DCB)	0.76	0.99	0.34	0.56	0.50	0.50
Patient’s initiatives (and companion) (PI)	0.76	0.98	0.61	0.44	0.58	0.42
Misalignments (MIS)	0.78	0.96	1.44	0.24	0.89	0.11
Interruptions (INT)	0.90	0.84	6.69	0.01	0.80	0.20
Accountability and expressions of trust in participants’ talk (ACC)	0.76	0.99	0.31	0.59	0.61	0.39
Markers of uncertainty in doctor’s talk (MOU)	0.79	0.95	1.85	0.18	0.90	0.10
Markers of emotions in doctor’s talk (MOE)	0.76	0.99	0.35	0.56	0.81	0.19

**Table 3 cancers-15-03008-t003:** Means for the presence of an accompanying person (Unaccompanied patient vs. accompanied patient).

Companion	Means							
	DCB	PI	MIS	INT	ACC	MOU	MOE	Valid N
Unaccompanied	8.63	4.58	3.63	0.54	1.33	0.33	1.63	24
Accompanied	9.17	4.78	3.00	1.11	.22	0.50	1.78	18
all Groups	8.86	4.67	3.36	0.79	1.29	0.41	1.69	42

Legend: DCB = doctor’s communicative behavior; PI = patient initiatives (and companion); MIS = misalignments; INT = interruptions; ACC = accountability and expressions of trust in participants’ talk; MOU = markers of uncertainty in the doctor’s talk; MOE = markers of emotions in doctor’s talk (MOE).

**Table 4 cancers-15-03008-t004:** Discriminant Function Analysis Summary for type of encounter (Follow-up vs. First Visit).

N = 42	Discriminant Function Analysis Summary No. of Vars in Model: 7; Grouping: Type of Encounter (2 Groups) Wilks’ Lambda: 0.61 Approx. F (7, 34) = 3.12 *p* < 0.05					
	Wilks’ Lambda	Partial Lambda	F-Remove (1, 34)	*p*-Level	Toler.	1-Toler (R-Sqr.)
Doctor’s communicative behavior (DCB)	0.65	0.93	2.69	0.11	0.56	0.44
Patient initiatives (and companion) (PI)	0.63	0.97	1.08	0.31	0.58	0.42
Misalignments (MIS)	0.62	0.98	0.81	0.37	0.90	0.10
Interruptions (INT)	0.66	0.92	3.18	0.08	0.77	0.23
Accountability and expressions of trust in participants’ talk (ACC)	0.65	0.93	2.66	0.11	0.71	0.29
Markers of uncertainty in doctor’s talk (MOU)	0.65	0.93	2.63	0.11	0.93	0.71
Markers of emotions in doctor’s talk (MOE)	0.63	0.96	1.48	0.23	0.79	0.21

**Table 5 cancers-15-03008-t005:** Standardized Coefficients for Canonical Variables.

ONCode Dimensions	Standardized Coefficients for Canonical Variables
	Root 1
Doctor’s communicative behavior (DCB)	0.58
Patient’s initiatives (and companion) (PI)	−0.37
Misalignments (MIS)	−0.26
Interruptions (INT)	0.53
Accountability and expressions of trust in participants’ talk (ACC)	0.51
Markers of uncertainty in doctor’s talk (MOU)	0.44
Markers of emotions in doctor’s talk (MOE)	−0.37

**Table 6 cancers-15-03008-t006:** Means for type of encounter (follow-up vs. first visit).

Type of Encounter	Means							
	DCB	PI	MIS	INT	ACC	MOU	MOE	Valid N
Follow-up	8.09	4.55	3.59	0.64	0.91	0.27	1.77	22
First Visit	9.70	4.80	3.10	0.95	1.70	0.55	1.60	20
all Groups	8.86	4.67	3.36	0.79	1.29	0.41	1.69	42

Legend: DCB = doctor’s communicative behavior; PI = patient’s initiatives (and companion); MIS = misalignments; INT = interruptions; ACC = accountability and expressions of trust in participants’ talk; MOU = markers of uncertainty in doctor’s talk; MOE = markers of emotions in doctor’s talk (MOE).

**Table 7 cancers-15-03008-t007:** Regression summary for dependent variable interruption.

N = 42	Regression Summary for Dependent Variable: Interruption R = 0.61 R^2^ = 0.37 Adjusted R^2^ = 0.26 F (6, 35) = 3.47 *p* < 0.01 Std. Error of Estimate: 0.62					
	Beta	Std. Err. of Beta	B	Std. Err. of B	t(35)	*p*-Level
Intercept			−1.21	0.83	−1.47	0.15
Type of Encounter	−0.05	0.15	−0.07	0.22	−0.34	0.73
Oncologist’s age	0.44	0.15	0.03	0.01	2.94	0.01
Patient’s age	−0.01	0.16	0.00	0.01	−0.04	0.97
Patient’s sex	0.05	0.15	0.08	0.24	0.32	0.75
Companion’s presence	0.49	0.15	0.70	0.21	3.30	0.00
Patient’s nationality	0.31	0.15	0.44	0.21	2.13	0.04

**Table 8 cancers-15-03008-t008:** Redundancy of independent variables and DV interruption.

Variable	Redundancy of Independent Variables; DV: Interruption The R-Square Column Contains the R-Square of the Respective Variable with All Other Independent Variables			
	Toleran.	R-Square	Partial Cor.	Semipart Cor
Type of Encounter	0.77	0.23	−0.06	−0.05
Oncologists’ age	0.82	0.19	0.45	0.39
Patients’ age	0.72	0.28	−0.01	−0.01
Patients’ gender	0.85	0.15	0.06	0.04
Companions’ presence	0.81	0.19	0.49	0.44
Patients’ nationality	0.85	0.15	0.34	0.29

**Table 9 cancers-15-03008-t009:** Semi-partial correlations between the VIs and the DV interruption.

Independent Variables	Semipartial Correlation	% Explained Variance
Type of Encounter	−0.05	0.21
Oncologists’ age	0.39	15.46
Patients’ age	−0.01	0.00
Patients’ gender	0.04	0.19
Companions’ presence	0.44	19.44
Patients’ Nationality (groups)	0.29	8.13

## Data Availability

The entire data corpus used in this study is not publicly available for privacy reasons. The collected video recordings of oncology encounters contain sensitive information that cannot be shared because it could compromise the privacy of the research participants. The data that support the findings of this study are available from the corresponding author, F.M., upon request.

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
