# Peer review of "Navigating Intercultural Medical Encounters: An Examination of Patient-Centered Communication Practices with Italian and Foreign Cancer Patients Living in Italy"

_cancers, 2023, doi:10.3390/cancers15113008_

Round 1
Reviewer 1 Report
Thank you for this important and interesting contribution to HCP-Patient communication, especially in the cross-cultural context for a country in Europe.
I would suggest to rephrase the paragraph and specify the countries for the references in the paragraph from lines 119 on. Your manuscript focuses on patients in Italy, but I assume references in this paragraph refer to minorities in the US and Canada? Also for 4. Discussion, line 385 referring to "more discrimination than white men in international studies", need to specify US (presumably).
The Simple Summary states "The study found no significant differences in communication between doctors and both Italian and foreign patients" (which I did find surprising), yet conclusions provide advice to pay attention to for foreign patients - "First, doctors should pay attention to the interruptions during visits, especially with foreign patients, and to the overall management of the unfolding interaction in intercultural encounters" and "our study discovered that even when foreign patients have enough linguistic competence, healthcare providers should not rely only on this element to assume good communication and quality care." The Simple Summary should be better aligned with the conclusions.
Congratulations on the article and interesting findings!
English is generally understood throughout, but need a final proofreading / spell- and grammar-check (e.g., 3.3 Qulitative analyzes) and rephrasing of some sentences.
Author Response
Reviewer 1.
I would suggest to rephrase the paragraph and specify the countries for the references in the paragraph from lines 119 on. Your manuscript focuses on patients in Italy, but I assume references in this paragraph refer to minorities in the US and Canada? Also for 4. Discussion, line 385 referring to "more discrimination than white men in international studies", need to specify US (presumably).
- Thank you for your valuable suggestions to enhance the clarity of our manuscript. We took your advice and have now specified the countries for the referenced studies in the concerned paragraphs. Indeed, the references from line 119 onward pertain to studies conducted in the US and Canada, focusing on ethnic minorities' experiences with PCC. In our discussion (line 385), we also highlighted that the noted discrimination experiences are primarily from international studies, predominantly based in the US. We sincerely hope these amendments address your concerns, and we appreciate your constructive feedback in improving our manuscript.
The Simple Summary states "The study found no significant differences in communication between doctors and both Italian and foreign patients" (which I did find surprising), yet conclusions provide advice to pay attention to for foreign patients - "First, doctors should pay attention to the interruptions during visits, especially with foreign patients, and to the overall management of the unfolding interaction in intercultural encounters" and "our study discovered that even when foreign patients have enough linguistic competence, healthcare providers should not rely only on this element to assume good communication and quality care." The Simple Summary should be better aligned with the conclusions.
- Thank you for your keen observation. We understand your concern regarding the perceived disparity between the Simple Summary and the Conclusions. Despite our findings indicating no significant differences in the overall communication patterns between doctors and both Italian and foreign patients, our study did identify certain areas for improvement in doctor-patient interactions, especially with foreign patients. We agree that the Simple Summary should accurately reflect these nuances. Following your recommendations, we have revised the Simple Summary to include the key message of our conclusion that: ' For instance, doctors should remain mindful of interruptions during visits, and they should foster patient-centered communication to ensure patient satisfaction and high-quality care. The language has also been simplified for a 14-year-old reading level. We trust that this revision of the simple summary more accurately aligns with the Conclusions. Once again, we deeply appreciate your constructive feedback and meticulous attention to detail.
Comments on the Quality of English Language
English is generally understood throughout, but need a final proofreading / spell- and grammar-check (e.g., 3.3 Qulitative analyzes) and rephrasing
- Following the reviewer's recommendation, we have had the manuscript proofread to ensure its adherence to the journal's standards. We have carefully reviewed and edited the entire manuscript to address any language and formatting issues, as well as to ensure the overall clarity and coherence of the content.
Reviewer 2 Report
Marino et al. examined the differences of patient-centered communication (PCC) practice between Italian and foreign cancer patients. Their analyses found no significant differences in communication between doctors and both Italian and foreign patients. I think the text is well-characterized, and the analysis generally makes sense. I only have a few minor comments.
1. 1. I think one of the flaws in this study is the sample size. For some weak factors, sample size 42 may not be sufficient to obtain significant P values. I think the authors may consider integrating their work with previously published studies. A meta-analysis, or a comparison of this study with previous results would strengthen their conclusions.
2. 2. The 20 foreigners are from 15 countries. Thus, I think a general statistical analysis may not be sufficient for such a study. Identifying and understanding certain outliers, such as scores outside the confidence interval, may be more useful for doctors dealing with emergency scenarios.
Author Response
Reviewer 2
- 1. I think one of the flaws in this study is the sample size. For some weak factors, sample size 42 may not be sufficient to obtain significant P values. I think the authors may consider integrating their work with previously published studies. A meta-analysis, or a comparison of this study with previous results would strengthen their conclusions.
- Thank you for your insightful comment regarding the sample size of our study. We appreciate your observation and agree that the sample size of 42 participants may be considered limited for certain weak factors, potentially affecting the attainment of significant P values. In response to your concern, we have included a statement in the manuscript acknowledging the limitation of the sample size as a primary constraint. Specifically, we added the following passage to address this limitation: "While the limited number of participants and the partial repetition of oncologists' com-munication characteristics and behaviors pose research limitations, this data set is valua-ble, despite the statistical constraint on the number of visits. Future research could lever-age larger data sets and ensure enough visits per oncologist. However, it is worth noting that the study's novelty and importance lie in its analysis of over 13,000 interaction turns from 18 hours of oncological visits, providing a rare view of the negotiation and co-construction of PCC in real intercultural interactions. Despite its complexity, such re-search into PCC as a multidimensional construct is scarce, underlining this study's value." We believe that this addition to the manuscript adequately addresses the concern raised regarding the sample size. While acknowledging the limitation, we emphasize the unique contribution of our study in analyzing a substantial number of interaction turns from a significant duration of oncological visits, shedding light on the negotiation and co-construction of patient-centered communication in real intercultural settings. We also acknowledge the need for future research to leverage larger data sets and ensure sufficient visits per oncologist.
- Thank you for your suggestion to integrate our work with previously published studies. We appreciate your input and recognize the value of connecting our findings with existing research in the field of patient-centered communication. In response to your recommendation, we have included additional studies in the discussion section of the manuscript. Specifically, we have incorporated new studies that highlight the positive impact of patient-centered communication on both native patients and foreigners, particularly those with a migrant background. We have added the following statement to address this point: "However, it is important to note that implementing PCC has the potential to bridge the racial and cultural differences between doctors and patients, and it can have a positive impact not only on both native and foreign patients. In this vein, our results align with previous research that has highlighted the positive effects of PCC on patients with a migrant background in the US 46,53,54. Paraphrasing Chu et al. 54 PCC “is key to reducing disparities and improving immigrant patients’ satisfaction level with medical care.” By including these additional studies, we aim to provide a broader context for our findings and establish connections with the existing literature on patient-centered communication. We appreciate your suggestion, as it has strengthened the discussion section of our manuscript and further enriched the overall contribution of our study.
- 2. The 20 foreigners are from 15 countries. Thus, I think a general statistical analysis may not be sufficient for such a study. Identifying and understanding certain outliers, such as scores outside the confidence interval, may be more useful for doctors dealing with emergency scenarios.
- Thank you for bringing up the point regarding the distribution of our foreign participants across 15 different countries. We certainly appreciate this perspective and understand its significance. We'd like to highlight that this diversity mirrors the demographics of Italy. As per the recent ISTAT census data (2022), foreign residents in Italy, who constitute 8.5% of the total population, show a significant degree of diversity. Romanians represent the most significant proportion, about 20%, with four other groups each comprising 5-10% and twelve more groups each making up 1-5%. Thus, the wide variety of nationalities in our sample is in alignment with the societal context in Italy, which we believe brings an added layer of authenticity to our findings. We look forward to further discussion and appreciate your valuable feedback.
- We greatly appreciate the reviewer's observation regarding potential outlier impacts, especially considering our sample size. In response, we conducted a thorough analysis of the outliers. Specifically, we recalculated the "Nationality" discriminant, which asked for the standardized discriminant scores of the two groups, Italians and Foreigners. This score is derived from the discriminant function, a linear combination of the seven ONCode predictors included in our analysis. It might have been reasonable to expect a greater dispersion among foreigners. However, interestingly, the dispersion was larger among Italians, with values ranging from -1.13 to 1.36 (SD = 0.73) for foreigners and -1.75 to 1.78 (SD = 1.19) for Italians. Within the confidence intervals, no outliers emerged among foreigners (Mean±SD = -1.12; 1.74), but we did identify one outlier within the Italian patients, with a discriminant score of -2.75 (Mean±SD = -2.62; 2.05). Wanting to thoroughly address the reviewer's insightful request to "understand certain outliers," we closely examined this patient's data. This patient is a female, unaccompanied, and older than the average age of the remaining Italian patients (70 vs. 60). Furthermore, her visit was characterized by no interruptions and a higher number of misalignments (12 versus 3.4 in the rest of the Italian sample). It's important to note that excluding this outlier from the discriminant did not change our results. We are very grateful for this opportunity to delve more deeply into our data, and we trust this addresses the reviewer's concern adequately.
Round 2
Reviewer 2 Report
The authors have addressed my concerns.